# Efficacy and Safety of Multi-Session Transcranial Direct Current Stimulation on Social Cognition in Schizophrenia: A Study Protocol for an Open-Label, Single-Arm Trial

**DOI:** 10.3390/jpm11040317

**Published:** 2021-04-19

**Authors:** Yuji Yamada, Takuma Inagawa, Yuma Yokoi, Aya Shirama, Kazuki Sueyoshi, Ayumu Wada, Naotsugu Hirabayashi, Hideki Oi, Tomiki Sumiyoshi

**Affiliations:** 1National Center of Neurology and Psychiatry, Department of Psychiatry, National Center Hospital, 4-1-1 Ogawahigashi-cho, Kodaira, Tokyo 187-8551, Japan; iki.witty.mind@gmail.com (Y.Y.); tinagawa@ncnp.go.jp (T.I.); yyokoi@ncnp.go.jp (Y.Y.); a.wada@ncnp.go.jp (A.W.); hirabaya@ncnp.go.jp (N.H.); 2National Center of Neurology and Psychiatry, Department of Preventive Intervention for Psychiatric Disorders, National Institute of Mental Health, 4-1-1 Ogawahigashi-cho, Kodaira, Tokyo 187-8551, Japan; shirama@ncnp.go.jp (A.S.); skzskz1000@gmail.com (K.S.); 3Translational Medical Center, National Center of Neurology and Psychiatry, Department of Clinical Epidemiology, 4-1-1 Ogawahigashi-cho, Kodaira, Tokyo 187-8551, Japan; oih@ncnp.go.jp

**Keywords:** neuromodulation, transcranial direct current stimulation (tDCS), schizophrenia, social cognition, superior temporal sulcus

## Abstract

Backgrounds: Social cognition is defined as the mental operations underlying social behavior. Patients with schizophrenia elicit impairments of social cognition, which is linked to poor real-world functional outcomes. In a previous study, transcranial direct current stimulation (tDCS) improved emotional recognition, a domain of social cognition, in patients with schizophrenia. However, since social cognition was only minimally improved by tDCS when administered on frontal brain areas, investigations on the effect of tDCS on other cortical sites more directly related to social cognition are needed. Therefore, we present a study protocol to determine whether multi-session tDCS on superior temporal sulcus (STS) would improve social cognition deficits of schizophrenia. Methods: This is an open-label, single-arm trial, whose objective is to investigate the efficacy and safety of multi-session tDCS over the left STS to improve social cognition in patients with schizophrenia. The primary outcome measure will be the Social Cognition Screening Questionnaire. Neurocognition, functional capacity, and psychotic symptoms will also be evaluated by the Brief Assessment of Cognition in Schizophrenia, UCSD Performance-Based Skills Assessment-Brief, and Positive and Negative Syndrome Scale, respectively. Data will be collected at baseline, and 4 weeks after the end of intervention. If social cognition is improved in patients with schizophrenia by tDCS based on this protocol, we may plan randomized controlled trial.

## 1. Introduction

The prevalence of schizophrenia is about 0.7% [1], with positive, negative, mood symptoms, and cognitive dysfunction. The first episode appears after adolescence and follows a chronic course with repeated remission and exacerbation of psychotic symptoms. In this process, cognitive and social function decline [2,3,4]. As a result, 70% of chronic patients are reported to be unable to find employment [5,6].

Cognitive dysfunction is one of the core symptoms of schizophrenia and exists from the early stage of onset to the chronic stage [7]. Several domains of neurocognition, such as learning memory, working memory, executive functioning, verbal fluency, and attention/information processing, are known to be impaired in schizophrenia [7]. Similarly, it has been noted in recent years that social cognition [8], i.e., mental operations underlying social behavior, is also impaired in schizophrenia. Social cognition consists of the domains of emotion recognition, social perception, theory of mind (ToM), and attributional bias [9], whose neural basis differs from that of neurocognition [10]. Moreover, it has been reported that improvement of social cognition is directly linked to improvement of social function, whereas there is argument that the association between neurocognition and social function is mediated by social cognitive function [11]. Therefore, in recent years, the importance of developing treatments for social cognition has been discussed.

The neural substrates of social cognition may include the orbitofrontal cortex, medial prefrontal cortex (mPFC), superior temporal sulcus (STS), and amygdala; these brain regions show a decrease in functional connectivity in schizophrenia [12]. Among these sites, the amygdala is involved in emotion recognition [13], while the prefrontal cortex governs ToM [14]. On the other hand, the STS is considered to play a role in both domains of social cognition [15] (see Table 1 [12,13,14,15,16,17,18,19]).

Neuromodulation is a method of changing nerve activity by applying electrical stimulation or the other agents to a specific nerve site in the body. Neuromodulation ranges from noninvasive approaches, such as transcranial magnetic stimulation, to invasive (implanted) approaches, such as spinal cord stimulation and deep brain stimulation [20]. Among them, transcranial direct current stimulation (tDCS) is a safe, inexpensive, and feasible neuromodulation that modifies nerve activity by providing a weak current of 1–2 mA for about 5–30 min/session [21]. The therapeutic effects of tDCS are thought to be mediated by promotion of cortical excitability through anodal stimulation [22]. In addition, enhanced excitatory synaptic transmission via anodal tDCS may promote glutamate transmissions, and suppress gamma-aminobutyric acid transmissions in the cortex. Moreover, it positively or negatively modulates the activities of dopamine, serotonin, and acetylcholine transmissions in the central nervous system. These neural events may change the balance between excitatory and inhibitory inputs [22,23]. With these mechanisms, tDCS has been suggested to ameliorate symptoms of several psychiatric disorders, e.g., major depressive disorder and schizophrenia.

tDCS has been shown to improve several domains of neurocognitive function, especially working memory, in schizophrenia [24]. On the other hand, only two studies have been conducted to determine the ability of tDCS to improve social cognitive disturbances of schizophrenia [25,26]. So far, only one study reported small facilitative effects of tDCS on emotion recognition with the left dorsolateral prefrontal cortex (DLPFC) as the target site in schizophrenia [25]. It should be noted that these studies used anodal stimulation on the frontal areas, i.e., the left DLPFC [26]. As social cognition may be governed mainly by other brain regions, e.g., the STS [12,13,14,15,16,17,18,19], it is hypothesized that anodal stimulation over this cortical area would be advantageous for treating social cognition disturbances.

These considerations prompt us to determine whether stimulation of skull surface above the STS, e.g., T3 or T4 (mid-temporal) of the International 10–20 electroencephalography system, would enhance social cognition in patients with schizophrenia. Moreover, intervention in the STS may affect hallucinations, as the STS is adjacent to the superior temporal gyrus (STG), and the network of cortical areas containing STGs is involved in hallucinations [27]. Therefore, we present a study protocol for an open-label, single-arm trial designed to evaluate the efficacy and safety of tDCS on the left STS. To our knowledge, this is the first attempt to administer tDCS targeting the STS in patients with schizophrenia.

## 2. Study Protocol

### 2.1. Trial Design

This study investigates the efficacy and safety of multi-session tDCS over the left STS to improve social cognition for patients with schizophrenia. This is a single-center trial at National Center of Neurology and Psychiatry, Tokyo, Japan. An open-label, single-arm study will be conducted on 15 participants with a diagnosis of schizophrenia based on the Diagnostic and Statistical Manual of Mental Disorders (DSM-5). We selected an open-label, single-arm design, because there is no precedent for tDCS over the left STS, and the major focus of this study is to verify the tolerability and safety of tDCS over the STS. Participants will receive 10 sessions of active tDCS in 5 consecutive days (twice per day) (see Figure 1). The study design is in accordance with the 2013 Standard Protocol Items: Recommendations for Interventional Trials (SPIRIT) Statement (Appendix A) [28]. This study was registered within the Japan Registry of Clinical Trials (Trial ID: jRCTs032180026).

### 2.2. Participants

Inpatients or outpatients treated at National Center Hospital, National Center of Neurology and Psychiatry will be enrolled. Participants will be recruited by referrals from treating psychiatrists. Those psychiatrists will not have any conflicts of interest with the outcomes of this trial. The principal investigator must provide written informed consent before starting the trial. After providing the informed consent, participants will be screened by a treating psychiatrist to establish whether they meet the eligibility criteria. Participants will be given a gift certificate worth JPY 3000 per day as a reimbursement, and a gift certificate worth JPY 21000 in a total of 7 days including each tDCS session, and baseline/follow-up evaluation.

### 2.3. Inclusion and Exclusion Criteria

Participants must meet the following inclusion criteria:(1)Diagnosed as schizophrenia in DSM-5.(2)Aged between 20 and 70.(3)Being able to understand the objectives and content of the study, and provide consent to participate in it. (The ability to consent to participate in this study will be considered insufficient, when patients’ Intelligence Quotient (IQ) is less than 70, or they present with acute psychiatric symptoms. Those patients will be provided with necessary medical care separately from this trial.)(4)Having Social Cognition Screening Questionnaire (SCSQ) scores of less than 34 points. Therefore, participants whose scores are 34 or more will be excluded from this study.

Patients with any of the following conditions will be excluded from the study:(1)Present or past history of severe organic lesions in the brain, dementia, or epilepsy.(2)With alcohol or substance use disorder that was present within 12 months from screening.(3)Contraindicated against electro convulsive therapy or tDCS, e.g., severe cardiovascular diseases, such as myocardial infarction, or aneurysms at high risk of rupture.(4)Were treated with tDCS or other neuromodulations within the past 2 months. (We will ask whether participants have any history of tDCS or other neuromodulations.)(5)Deemed inappropriate to participate judged by the principal investigator, e.g., when participants’ psychiatric symptoms are unstable.

The dose of psychotropic drugs will not be changed during the study period. Cognitive rehabilitation will not be performed during the period. Therefore, we will exclude patients who are scheduled for cognitive rehabilitation.

### 2.4. Sample Size Calculation

Total study sample sizes of *n* = 15 have been recommended by our previous study [29], assuming an estimated mean UCSD Performance-based Skills Assessment-brief (UPSA-B) difference from baseline to follow-up of 10.6, with a standard deviation of 15.5. Under these conditions, the power of the primary analysis was 0.8, so approximately *n* = 13 was estimated (one-sample Student’s t-test). Therefore, it was decided to include a total of 15 samples, taking into account the dropouts of the study.

### 2.5. Intervention

Direct current will be transmitted through 35 cm^2^ saline-soaked sponge electrodes, and the intervention will be performed by a 1 × 1 transcranial direct current low-intensity stimulator (Model 1300 A; Soterix Medical Inc., New York, NY, USA). According to the International 10–20 electroencephalography system, in each session, the tDCS montage will place the anode in the left STS and the cathode in the contralateral supraorbital region, which corresponds to the T3 (mid-temporal) and FP2 (front-polar) regions (see Figure 2). We will apply 10 sessions of direct current of 2 mA for 20 min in 5 consecutive days (twice per day, with an interval of 30 min). The intensity, frequency, and duration of the stimulus are determined based on previous studies [29].

Trained psychiatrists or researchers will administer tDCS, and they will not evaluate any outcome measures. Neither tDCS-administrants nor participants will be aware of their treatment results until all participants have finished their follow-up evaluations.

### 2.6. Outcomes

Patients receive a psychological evaluation, including a screening evaluation, after being briefed on the purpose of the study and agreeing to participate in the study. Psychological assessment data will be collected at baseline and 4 weeks after the final stimulus (see Table 2). Baseline and follow-up evaluations will be performed by experienced psychologists who are not blinded. Although the interval between the baseline and follow-up evaluation is more than 5 weeks, which secures adequate washout period, the leaning effect of psychological evaluation will be a possible limitation.

#### 2.6.1. Cognition

The primary outcome is scores on the Social Cognition Screening Questionnaire (SCSQ) [30], which includes test of attributional style, and theory of mind (ToM). To evaluate ToM more accurately, we will also use False Belief Task [31], Hinting Task [32], and Autism-Spectrum Quotient (AQ) [33]. To evaluate emotion recognition, we will use the Facial Emotion Selection Test (FEST) [34]. To provide a standard metric for combining test scores into domains and comparing performance over time, Brief Assessment of Cognition in Schizophrenia (BACS) scores will be converted to z-scores, which shows performance relative to healthy people [35]. The premorbid IQ will be also estimated using the Japanese Adult Reading Test (JART) [36] (see Table 3).

#### 2.6.2. Functional Capacity (Daily-Living Skills)

The functional capacity will be assessed by the UCSD Performance-Based Skills Assessment-Brief (UPSA-B) [37], which consists of financial and communication skills.

#### 2.6.3. Global Symptoms of Schizophrenia

Global symptoms of schizophrenia will be evaluated by the Positive and Negative Syndrome Scale (PANSS) [38], which consists of positive syndrome, negative syndrome, and general psychopathology subscales.

#### 2.6.4. Adverse Events

Adverse events are defined as unwanted experiences seen during tDCS. Serious adverse events are defined as requiring inpatient treatment, moderate adverse events as requiring therapeutic intervention, and mild adverse events as requiring no therapeutic intervention. The treating physician will record the symptoms, date of onset, severity, treatment given, and association with research interventions. If symptoms are already present at baseline and do not worsen during tDCS intervention, they are not treated as adverse events.

All adverse events will be clinically evaluated and monitored throughout the study period. Previous studies report that the most common adverse events are itching, tingling, headache, burning sensation, and discomfort [39]. An experienced psychiatrist will check the presence and extent of adverse events and their association with tDCS before and after each session and assess safety at all visits during the intervention. We will follow up any unresolved adverse events after trial completion. The principal investigator will be responsible for addressing and explaining serious adverse events in the patient. The sub-investigators will be responsible for reporting any information related to such adverse events to the principal investigator. The principal investigator will have to report any serious adverse events to the Clinical Research Review Board, and to the Ministry of Health, Labor and Welfare. We will cease intervention on a per-patient if we observe severe adverse events and will also cease the whole study if we observe severe adverse events in two patients.

#### 2.6.5. Prescribed Drugs

We will collect information about prescribed drugs throughout the study period. Those drugs will be classified, in principle, into the four categories: antipsychotics, mood stabilizers, antidepressants, and benzodiazepines. Furthermore, the equivalent doses of chlorpromazine, diazepam, and imipramine will be calculated [40].

### 2.7. Data Collection and Data Management

The assessments will be conducted at baseline and 4 weeks after the end of the last stimulation (Table 2). All evaluations will be conducted by experienced psychologists. The data will initially be recorded in a paper file and each participant will be assigned a code number. These files will be stored in a locked security box. After the follow-up data are collected, all data in the paper files will be transcribed to the Electronic Data Capture system (HOPE eACReSS; Fujitsu, Tokyo, Japan), which is a secure system designed for storage of personal and patient data. The data will be sent to independent data managers to assess whether the data are collected properly, focusing on the status of consent acquisition, eligibility of participants, evaluation items, and confirmation of drop-out/terminated cases. These data managers will also oversee and review the progress of the trial. If a participant withdraws their consent, they will be dismissed from the study. At the same time, we will record the dropout rate, and the number of people with adverse events requiring treatment. We will define a dropout rate of less than 10% as a safety criterion and perform a quantitative assessment of safety. The Efficacy and Safety Assessment Committee, whose members are independent of the research and come from the National Center of Neurology and Psychiatry, will check and assess whether the trial is conducted safely and properly, and will also decide whether to stop the trial if any severe adverse events or protocol violations occur. In addition, an on-site data monitor will conduct monitoring to ensure the trial is performed properly, data is properly recorded, and data reliability is ensured. If we conduct any necessary protocol modifications, we will report them to the Clinical Research Review Board, and to the Ministry of Health, Labor and Welfare for registration in the Japan Registry of Clinical Trials website (https://jrct.niph.go.jp).

### 2.8. Statistical Analysis

Correlations between baseline values and their changes from baseline of SCSQ, Hinting Task, FEST, BACS, UPSA-B, and PANSS scores, will be evaluated. Correlations will be examined for chlorpromazine equivalent dose of antipsychotics vs. changes from baseline of SCSQ, Hinting Task, FEST, BACS, UPSA-B, and PANSS scores. Correlations will also be examined for the change of UPSA-B scores from baseline and changes of the corresponding scores from baseline.

Statistical analysis will be conducted using STATA 14, created by StataCorp in TX, USA. For continuous variables in the SCSQ, Hinting Task, FEST, BACS, UPSA-B, and PANSS, we will use Student’s t-test. Pearson’s product moment correlation coefficient will be used for the relationship between clinical variables.

## 3. Ethics Statement

The study will be performed according to the Declaration of Helsinki and will follow the Clinical Trials Act in Japan. The protocol has been presented for approval by the National Center of Neurology and Psychiatry Clinical Research Review Board (CRB3180006). The principal investigator (TS) will have the ultimate responsibility for providing informed consent to all research participants. All participants must agree to participate in the study. After initial review and approval (September 2018), the institution’s clinical research review committee will review the protocol and implementation at least annually. Two annual reviews have proceeded to date (https://jrct.niph.go.jp/re/bulletins/detail/312/22, https://jrct.niph.go.jp/re/bulletins/detail/312/1540). The principal investigator should submit a safety and progress report to the Review Board at least annually, and the researcher should submit the final report within 3 months of the completion of the study. These reports will include a summary of the total number of enrolled participants, serious/nonserious adverse events that occurred, and safety and monitoring committee reviews [41]. Changes related to the research protocol need to be submitted to the Clinical Research Review Board for review as a protocol modification. If a participant needs to be treated due to a moderate or severe adverse event directly caused by tDCS, the participant will receive the full cost of treatment from clinical trial insurance.

## Figures and Tables

**Figure 1 jpm-11-00317-f001:**
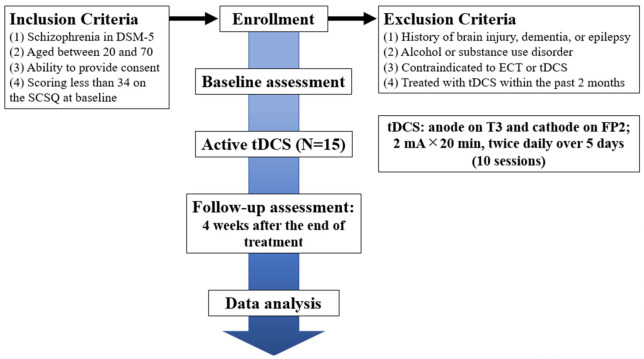
Flowchart summarizing the trial procedure. SCSQ, Social Cognition Screening Questionnaire; ECT, electroconvulsive therapy; tDCS, transcranial direct current stimulation.

**Figure 2 jpm-11-00317-f002:**
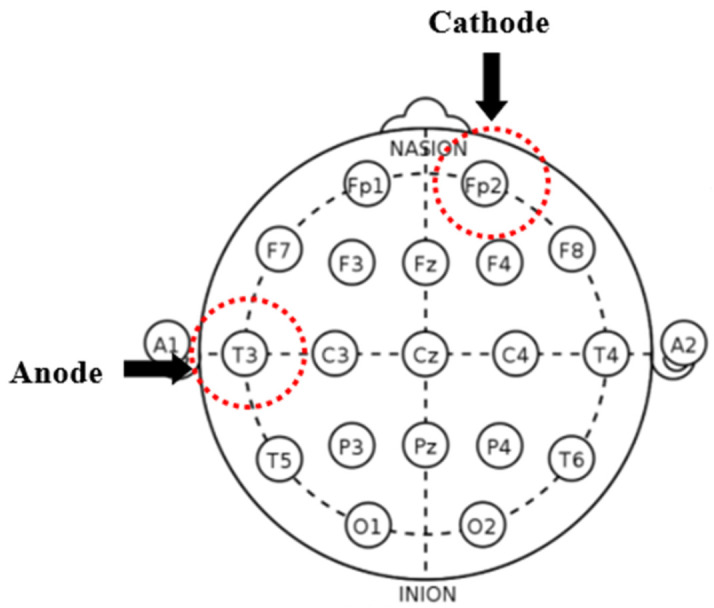
Placement map for the intervention. tDCS montage will place the anode in the left superior temporal sulcus (STS) and the cathode in the contralateral supraorbital region, which corresponds to the T3 (mid-temporal) and FP2 (front-polar) regions in the International 10–20 electroencephalography system.

**Table 1 jpm-11-00317-t001:** Neural basis of social cognition [12,13,14,15,16,17,18,19].

Domains of Social Cognition	Neural Basis
Emotion recognition	Amygdala, superior temporal sulcus, medial prefrontal cortex, inferior occipital gyrus, etc.
Theory of mind (ToM)	Superior temporal sulcus, medial prefrontal cortex, middle temporal gyrus, etc.
Attributional bias	Orbitofrontal cortex, insular cortex, striatum, amygdala, superior temporal sulcus, etc.

**Table 2 jpm-11-00317-t002:** Study schedules.

	Study Period				
	Baseline	Intervention			Follow-Up
**Time point**	Within 2 weeks before the start of intervention	Day 1	Days 2–4	Day 5	4 weeks after the end of the last stimulation
**Enrollment**					
Eligibility screen	X				
Informed consent	X				
Sociodemographic characteristics	X				
**Intervention**					
tDCS (twice/day)		X	X	X	
Assessments					
SCSQ	X				X
Hinting Task	X				X
FEST	X				X
False Belief Task	X				X
BACS	X				X
UPSA-B	X				X
PANSS	X				X
AQ	X				X
JART	X				
Adverse events	X	X	X	X	X
Prescribed drugs	X	X	X	X	X

tDCS, transcranial direct current stimulation; SCSQ, Social Cognition Screening Questionnaire; FEST, Facial Emotion Selection Test; BACS, Brief Assessment of Cognition in Schizophrenia; UPSA-B, Brief UCSD Performance-based Skills Assessment; PAMSS, Positive and Negative Syndrome Scale; AQ, Autism-Spectrum Quotient; JART, Japanese Adult Reading Test. The timepoint of follow-up evaluation will be allowed to be up to 7 days off.

**Table 3 jpm-11-00317-t003:** Primary/secondary outcomes.

Primary/Secondary	Domain	Outcome
Primary outcome	Social cognition (attributional style, theory of mind)	Social Cognition Screening Questionnaire (SCSQ)
Secondary outcome	Social cognition (theory of mind)	Hinting Task
Secondary outcome	Social cognition (theory of mind)	False Belief Task
Secondary outcome	Social cognition (emotion recognition)	Facial Emotion Selection Test (FEST)
Secondary outcome	Cognition	Brief Assessment of Cognition in Schizophrenia (BACS)
Secondary outcome	Cognition	Autism-Spectrum Quotient (AQ)
Secondary outcome	Premorbid intelligence quotient	Japanese Adult Reading Test (JART)
Secondary outcome	Functional capacity (daily-living skills)	UCSD Performance-Based Skills Assessment-Brief (UPSA-B)
Secondary outcome	Global symptoms of schizophrenia	Positive and Negative Syndrome Scale (PANSS)

## Data Availability

We registered the protocol information in the Japan Registry of Clinical Trials website (https://jrct.niph.go.jp).

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
