# Peer review of "Efficacy and Safety of Multi-Session Transcranial Direct Current Stimulation on Social Cognition in Schizophrenia: A Study Protocol for an Open-Label, Single-Arm Trial"

_jpm, 2021, doi:10.3390/jpm11040317_

Round 1
Reviewer 1 Report
The authors have addressed previous review comments and revised the manuscript appropriately.
Minor recommendations:
Section 2.6.4: “… will also cease the whole study if we observe severe adverse events in multiple patients.” What number of patients (absolute number or proportion of those recruited) experiencing severe adverse events will result in the authors stopping the study?
Section 2.4: The authors use a Student’s t-test to evaluate change from baseline to follow up and predict power. Did the authors use a paired sample t-test as appropriate in this study design?
Author Response
April 6, 2021
Dear Editor,
Thank you very much for your correspondence regarding our paper titled “Efficacy and safety of multi-session transcranial direct current stimulation on social cognition in schizophrenia: A study protocol for an open-label, single-arm trial”.
Based on the reviewers’ comments, the manuscript has been revised with additional information, as follows.
Reviewer 1
Section 2.6.4
Comment 1: “… will also cease the whole study if we observe severe adverse events in multiple patients.” What number of patients (absolute number or proportion of those recruited) experiencing severe adverse events will result in the authors stopping the study?
Response 1: In view of the Reviewer’s comment, we have revised the relevant sentences, as follows;
“We will cease intervention on a per-patient if we observe severe adverse events, and will also cease the whole study if we observe severe adverse events in 2 patients.”
Section 2.4
Comment 2: The authors use a Student’s t-test to evaluate change from baseline to follow up and predict power. Did the authors use a paired sample t-test as appropriate in this study design?
Response 2: In view of the Reviewer’s comment, we have revised the relevant sentences, as follows;
“Under these conditions, a power of the primary analysis was 0.8, so approximately n=13 was estimated (one-sample Student's t-test).”
We believe that we have addressed all of the issues raised by the reviewer, and hope you will find this revision acceptable for publication in your Journal.
Sincerely
Tomiki Sumiyoshi, MD, PhD
Department of Preventive Intervention for Psychiatric Disorders,
National Institute of Mental Health, National Center of Neurology and Psychiatry,
4-1-1 Ogawahigashi-cho, Kodaira, Tokyo 187-8551, Japan.
Telephone: +81-42-341-2711, FAX: +81-42-344-6745
E-Mail: sumiyot@ncnp.go.jp

Reviewer 2 Report
Dear Authors,
Thank you for your time.
This protocol is quite well organized for studying tDCS therapy for schizophrenia.
However, there are still several problems that should be addressed.
- In 2013, Sri Mahavir Agarwal et al. have reviewed the articles on tDCS therapy for schizophrenia and indicated its effectiveness in releasing symptoms [1]. Similarly, tDCS was demonstrated to improve different aspects of cognition levels in schizophrenia patients; some studies have reported differential benefits of tDCS (versus sham) interventions on trained cognitive tasks [2-4]. Why the authors prefer the single-arm study instead of randomized controlled trial (RCT)?
- The tDCS has been applied to test the left posterior superior temporal gyrus (STG) involvement in creating false auditory perceptions. It showed that tDCS increased excitability levels in STG, leading to a higher rate of ‘false alarm’ [5]. This could have an impact on the schizophrenia patients’ reaction or manifestations.
- The interaction factors between drugs and tDCS is worthy attention to clarify the origins of the clinical improvement. Which can be added in the statistical analysis.
Reference:
- Agarwal SM, Shivakumar V, Bose A, Subramaniam A, Nawani H, Chhabra H, Kalmady SV, Narayanaswamy JC, Venkatasubramanian G (2013) Transcranial direct current stimulation in schizophrenia. Clin Psychopharmacol Neurosci 11: 118-125 doi:10.9758/cpn.2013.11.3.118
- Jahshan C, Rassovsky Y, Green MF (2017) Enhancing Neuroplasticity to Augment Cognitive Remediation in Schizophrenia. Front Psychiatry 8: 191 doi:10.3389/fpsyt.2017.00191
- Green MF, Horan WP, Lee J (2019) Nonsocial and social cognition in schizophrenia: current evidence and future directions. World Psychiatry 18: 146-161 doi:10.1002/wps.20624
- Gupta T, Kelley NJ, Pelletier-Baldelli A, Mittal VA (2018) Transcranial Direct Current Stimulation, Symptomatology, and Cognition in Psychosis: A Qualitative Review. Front Behav Neurosci 12: 94 doi:10.3389/fnbeh.2018.00094
- Moseley P, Fernyhough C, Ellison A (2014) The role of the superior temporal lobe in auditory false perceptions: a transcranial direct current stimulation study. Neuropsychologia 62: 202-208 doi:10.1016/j.neuropsychologia.2014.07.032
Author Response
April 6, 2021
Dear Editor,
Thank you very much for your correspondence regarding our paper titled “Efficacy and safety of multi-session transcranial direct current stimulation on social cognition in schizophrenia: A study protocol for an open-label, single-arm trial”.
Based on the reviewers’ comments, the manuscript has been revised with additional information, as follows.
Reviewer 2
Comment 3: In 2013, Sri Mahavir Agarwal et al. have reviewed the articles on tDCS therapy for schizophrenia and indicated its effectiveness in releasing symptoms [1]. Similarly, tDCS was demonstrated to improve different aspects of cognition levels in schizophrenia patients; some studies have reported differential benefits of tDCS (versus sham) interventions on trained cognitive tasks [2-4]. Why the authors prefer the single-arm study instead of randomized controlled trial (RCT)?
Response 3: We have provided the explanation on this issue in the manuscript, as follows;
“We selected an open-label, single-arm design, because there is no precedent for tDCS over the left STS, and the major focus of this study is to verify the tolerability and safety of tDCS over the STS.”
Comment 4: The tDCS has been applied to test the left posterior superior temporal gyrus (STG) involvement in creating false auditory perceptions. It showed that tDCS increased excitability levels in STG, leading to a higher rate of ‘false alarm’ [5]. This could have an impact on the schizophrenia patients’ reaction or manifestations.
Response 4: We appreciate the comment by the reviewer, and have added the relevant sentences to the manuscript, as follows;
“Moreover, intervention in the STS may affect hallucinations, as the STS is adjacent to the superior temporal gyrus (STG), and the network of cortical areas containing STGs is involved in hallucinations [41].”
Comment 5: The interaction factors between drugs and tDCS is worthy attention to clarify the origins of the clinical improvement. Which can be added in the statistical analysis.
Response 5: We have provided the explanation on this issue in the manuscript, as follows;
“Correlations will be examined for chlorpromazine equivalent dose of antipsychotics vs. changes from baseline of SCSQ, Hinting Task, FEST, BACS, UPSA-B, and PANSS scores.”
Reference:
- Agarwal SM, Shivakumar V, Bose A, Subramaniam A, Nawani H, Chhabra H, Kalmady SV, Narayanaswamy JC, Venkatasubramanian G (2013) Transcranial direct current stimulation in schizophrenia. Clin Psychopharmacol Neurosci 11: 118-125 doi:10.9758/cpn.2013.11.3.118
- Jahshan C, Rassovsky Y, Green MF (2017) Enhancing Neuroplasticity to Augment Cognitive Remediation in Schizophrenia. Front Psychiatry 8: 191 doi:10.3389/fpsyt.2017.00191
- Green MF, Horan WP, Lee J (2019) Nonsocial and social cognition in schizophrenia: current evidence and future directions. World Psychiatry 18: 146-161 doi:10.1002/wps.20624
- Gupta T, Kelley NJ, Pelletier-Baldelli A, Mittal VA (2018) Transcranial Direct Current Stimulation, Symptomatology, and Cognition in Psychosis: A Qualitative Review. Front Behav Neurosci 12: 94 doi:10.3389/fnbeh.2018.00094
- Moseley P, Fernyhough C, Ellison A (2014) The role of the superior temporal lobe in auditory false perceptions: a transcranial direct current stimulation study. Neuropsychologia 62: 202-208 doi:10.1016/j.neuropsychologia.2014.07.032
We believe that we have addressed all of the issues raised by the reviewer, and hope you will find this revision acceptable for publication in your Journal.
Sincerely
Tomiki Sumiyoshi, MD, PhD
Department of Preventive Intervention for Psychiatric Disorders,
National Institute of Mental Health, National Center of Neurology and Psychiatry,
4-1-1 Ogawahigashi-cho, Kodaira, Tokyo 187-8551, Japan.
Telephone: +81-42-341-2711, FAX: +81-42-344-6745
E-Mail: sumiyot@ncnp.go.jp

Round 2
Reviewer 2 Report
Thanks for the author's answer. I wish you success in your research.
This manuscript is a resubmission of an earlier submission. The following is a list of the peer review reports and author responses from that submission.
Round 1
Reviewer 1 Report
The authors report on a study protocol for the investigation of tDCS in schizophrenia. The study is already approved by the local review board and registered with the relevant national trials registry (Trial ID: jRCTs032180026), and appears to have begun recruiting participants. No results have been reported on the clinical trials registry to date.
Major recommendations:
I note overlapping authors have been involved in a similar study https://pubmed.ncbi.nlm.nih.gov/33361162/ which reports a similar intervention with different study design. I suggest the authors consider the level of detail with which this other protocol is written and aim to provide a similar level of detail here. I give examples of where additional detail is appropriate in the study design and limitations which must be discussed.
Section 2.1: Can the authors please justify the experimental design and comment on the limitations of this design. Why was an unblinded, uncontrolled design proposed?
Section 2.6: Does the learning effect present a confound for the study design? The authors propose to repeat assessment with a series of tests and questionnaires. Would subjects be expected to change results if these tests are administered in multiple stages?
Section 2.6.1 to 2.6.3: Compared with the registered trial the following secondary outcome is removed “JART25 (Japanese Adult Reading Test-25)”, and the following secondary outcome is added “Hinting Task”. Can the authors clarify this discrepancy?
The authors should consider what supplementary materials are appropriate. For example redacted Patient Information Leaflets and Consent forms, as recommended in SPIRIT guidelines.
Minor recommendations:
Page 2, section 1, paragraph 1: “[5.6]” typo, should be a comma.
Section 1. References appear to be misordered. Jumps from [11] to [19].
Section 1. It appears that studies [17] and [18] are miscited. “… in schizophrenia with limited effects [17, 18]. It should be noted that the majority of the previous study [18] used anodal stimulation on the frontal areas…” Is the previous study not [17]? [18] is a systematic review. The authors should give a brief summary of results from this study.
Section 2.2: Is any incentive or reimbursement given for participants to take part?
Section 2.2: Are referring psychiatrists involved in, or have conflict on interest with, the outcomes of the study? Or are these referrals independent of the study team? This appears to be a possible source of sample bias.
Section 2.3: How will the authors determine whether participants are able to understand the study and provide consent? Will impartial assistance be provided to participants who could be considered vulnerable?
Section 2.3: If a participant is included, and scores >34 on SCSQ in the baseline, is this participant then removed from the study?
Section 2.3: Give examples of what constitutes “contraindicated against ECT or tDCS”.
Section 2.3: Please give examples of what constitutes participants “deemed inappropriate to participate judged by the principal investigator”? This appears to be a possible source of sample bias.
Section 2.3: Did the authors consider an inclusion criterion of previous tDCS treatment with no observed adverse effects? Will the authors ask whether participants have any history of tDCS treatment?
Section 2.3: “Cognitive rehabilitation will not be performed during the period.” Will this comprise a break in treatment for patients?
Section 2.4: For which statistical method was the power calculated?
Section 2.6: Will the researchers or psychologists evaluating the outcomes be blinded to whether the outcome is from baseline or post-treatment?
Section 2.5: Please provide a placement map for the intervention.
2.6.1 to 2.6.3: Please provide a clear table, or list, of primary and secondary outcomes.
Section 2.6.4: “The principal investigator will be responsible for explaining any serious adverse events in the relevant patients.” To whom does the principal investigator explain these adverse events? To the participants? To the local and national bodies? What constitutes a “serious” adverse event compared with all events which will be recorded? Is there a required timescale for reporting adverse events: to the PI; and from the PI to the local and national bodies?
Section 2.6.4: Are the “Clinical Research Review Board” and “Ministry of Health, Labor and Welfare” separate bodies, or the same body? Please clarify. e.g. "the Clinical Research Review Board and the Ministry of Health, Labor and Welfare..." or "the Clinical Research Review Board at the Ministry of Health, Labor and Welfare...".
Section 2.7: Is HOPE eACReSS a secure system designed for storage of personal and patient data?
Section 2.7: What assessments will the independent data managers carry out to assess whether data is collected properly?
Section 2.7: What will happen to a participant’s already-collected data if they withdraw from the study? Will the authors record and report the number of withdrawals and study stage?
Section 2.7: “We will also cease intervention if we observe severe adverse events.” Is this on a per-patient? Or whole study basis? It appears this statement should be moved to section 2.6.4.
Table 2: Is the follow up date exactly 4 weeks after day 5 (i.e. day 33)? Or is some variation allowed? If so, how much variation and will the authors account for this in their analysis?
Throughout: I suggest the authors use the term “participants” rather than “subjects” throughout the manuscript.
Section 3. Given initial approval was September 2018, and annual review is planned, can the authors confirm that 2 annual reviews have proceeded to date?
Reviewer 2 Report
This protocol applied different mental measurement scales for evaluation on the tDCS on cognition in schizophrenia.
The article needs to be polished by native speakers.
Major problems:
- When the enrolled patients take drugs, whether there are any interaction effects between the drug and tDCS, which is the primary factor in the improvement? Please list the protocol for collecting those data and analysis methods.
- Yuri Rassovsky has already published a study using tDCS on social cognition in schizophrenia in 2015 (Rassovsky et al., 2015). What is the speciality for the current protocol?
- Line 157 “twice per day, with an interval of 30 minutes or more”. Please explain why the time interval is not fixed at 30 min? Will the longer interval impact the efficacy of tDCS? Monte-Silva suggests that the stimulation interval might be necessary for the aftereffects of cathodal tDCS (Monte-Silva, Kuo, Liebetanz, Paulus, & Nitsche, 2010).
- In 2020, your team published a meta-analysis titled “Effect of multi-session prefrontal transcranial direct current stimulation on cognition in schizophrenia: A systematic review and meta-analysis” (Narita et al., 2020). The tDCS is placed on the T3 or F3 in the enrolled studies, and the anode was mostly on the F3. Why is it on the T3 in the current protocol?
Minor problems:
- Line 156 “T3 and FP2 regions”. Please use the full name to interpret them.
- Line 179 ~ 180 “The functional capacity will be assessed by the UCSD Performance-based Skills Assessment-brief (UPSA-B), which consists of finance and communication subscales”. It could be changed to “The functional capacity will be assessed by the UCSD Performance-based Skills Assessment-brief (UPSA-B), which consists of subscales financial and communication skills”.
- Why the Autism-Spectrum Quotient (AQ) and Japanese Adult Reading Test (JART) are not tested in the follow-up phase?
Reference
Monte-Silva, K., Kuo, M. F., Liebetanz, D., Paulus, W., & Nitsche, M. A. (2010). Shaping the optimal repetition interval for cathodal transcranial direct current stimulation (tDCS). J Neurophysiol, 103(4), 1735-1740. doi:10.1152/jn.00924.2009
Narita, Z., Stickley, A., DeVylder, J., Yokoi, Y., Inagawa, T., Yamada, Y., . . . Sumiyoshi, T. (2020). Effect of multi-session prefrontal transcranial direct current stimulation on cognition in schizophrenia: A systematic review and meta-analysis. Schizophr Res, 216, 367-373. doi:10.1016/j.schres.2019.11.011
Rassovsky, Y., Dunn, W., Wynn, J., Wu, A. D., Iacoboni, M., Hellemann, G., & Green, M. F. (2015). The effect of transcranial direct current stimulation on social cognition in schizophrenia: A preliminary study. Schizophr Res, 165(2-3), 171-174. doi:10.1016/j.schres.2015.04.016